# Factor Structure and Psychometric Properties of Emotional Eater Questionnaire (EEQ) in Spanish Colleges

**DOI:** 10.3390/ijerph17239090

**Published:** 2020-12-05

**Authors:** Elena Bernabéu, Carlos Marchena, María Teresa Iglesias

**Affiliations:** 1Faculty of Education and Psychology, Universidad Francisco de Vitoria, 28223 Madrid, Spain; 2Faculty of Health Sciences, Universidad Francisco de Vitoria, 28223 Madrid, Spain; m.iglesias.prof@ufv.es

**Keywords:** emotional eater, EEQ, EFA, factor, psychometric, college students

## Abstract

Emotional eating (EE) patterns have been shown to play a relevant role in the development of overweight problems. However, there is a gap in research aimed at validating questionnaires to assess EE in specific populations. The aim of the study was to analyze factor structure and psychometric properties of Emotional Eater Questionnaire (EEQ) in Spanish universities. EEQ, state-anxiety subscale of STAI and a questionnaire about health habits were filled out by 295 students. Exploratory Factor Analysis (EFA) by using Unweight Least Squares (ULS) method was carried out. To determine factor numbers we used eigenvalues, parallel analysis, and goodness of fit statistics. Cronbach’s alpha and Spearman correlations were used to analyze reliability, convergent, and concurrent validity. The parallel analysis and goodness of fit statistics showed that unifactorial structure of seven items was the most appropriate what accounted for 57% of the variance. Internal consistency was good (α = 0.753), as well as convergent validity (r = 0.317; *p* < 0.001). Concurrent validity was significant for three of the five criteria (r = −0.224; *p* < 0.001 and r = −0.259; *p* < 0.001). The results suggest some differences in the structure of the psychometric assessment of EE in sub-clinical population in comparison with previous studies carried on with an overweight population, what could be relevant to obesity prevention.

## 1. Introduction

Emotional eating (EE) refers to eating that is triggered by emotions [1] and has been originally defined as the tendency to consume food, or to increase food intake, in order to cope with negative emotional states [2,3]. However, a number of studies show that a positive mood can also elicit increased food intake in emotional eaters [4], and several researchers have accepted positive emotions as part of emotional eating [5]. Among several different mechanisms, deficits in executive function, such as reduced inhibitory control or impulsivity, are associated with EE [6,7]. Closely related, emotion regulation difficulties (control of the modulation of arousal, but also the awareness and understanding of emotions) have been significantly related to EE [7,8,9,10,11].

Episodes of EE usually include the intake of a large amount of calorie dense, sweet and/or high fat foods [12,13]. EE has been associated with increased carbohydrate intake for women, and, in men, fat-free mass content was associated with lower scores in EE [14]. In a non-convergent way, although, it has also been found the lack of significant effects of EE on caloric intake [15] or that EE status did not predict the consumption of a specific food type [16].

Prevalence estimate varies according to different studies, with approximately 20–45% of non-clinical adult samples identifying as individuals who engage in emotional eating [17,18]. Among adults with overweight rates are closer to 60% [19]. Although obesity is considered the result of a variety of interactions between several factors (genetic, socioeconomic, endocrine, metabolic and psychological) [20], empirical research suggests that EE is quite common among individuals with excess weight, who score higher on measures of EE than those individuals within the “normal” weight category [21,22]. EE affects more than half of people referred for obesity treatment [23]. In addition, EE has been positively related to BMI [12], and has been regarded as a risky behavior for obesity [24], and as a strong psychological predictor of weight gain [25], even in children and adolescents [26]. The effect of overweight on health is clear: adults with obesity have a 20% or higher risk of dying of all-cause mortality compared with adults of normal weight [27], so it’s necessary to consider all the factors involved in excess weight, as EE and the mechanisms involved, to improve prevention programs regarding obesity. The treatment of people with obesity and high EE should not only focus on calorie-restricted diets, but on regulation skills [28]. It is clear the need of the implementation of treatment interventions that address emotion regulation and include strategies to cope with impulsivity [7].

Although EE can be associated with positive emotions, these episodes often seems an attempt to avoid, control, or cope with negative emotions, like anxiety, sadness or anger [1]. Anxiety has been implicated as a primary trigger for EE episodes [29], and it has been found that trait anxiety (but not trait anger) may be useful for predicting EE in individuals with obesity [30], in line with many previous studies that have shown the potential effect of anxiety on eating behavior [31,32]. There is also robust evidence that EE is a mediator between depression and obesity [28].

University students are in a critical transition period towards adulthood, and the university experience is usually accompanied by significant changes, as demand for greater autonomy, more self-discipline in daily routines and stress experienced [33]. Emotional distress seems to be the principal psychological problem experienced during university studies [34,35]. The worry about academic performance appears to be the main driver of this distress [36,37]. The absence of effective strategies to deal with situations of stress may result in the adoption of inappropriate coping mechanisms, such unhealthy or harmful eating habits emotional as EE [38]. A recent study suggests that the grade point average, academic self-esteem, and academic worries in adolescences were related to emotional eating scores [39].

Research on EE, the mechanisms involved and its clinical correlates, requires the development and validation of psychometric scales to identify and quantify this construct. Several instruments have been used in the study of EE. One of these questionnaires is the Emotional Eating Scale (EES) [1] and the revised version (EES-R) [40], designed to measure the urge to eat in response to specific negative emotional states and overeating. The EES-R, 25-item scale, includes three subscales of EE behavior: depression EE, boredom EE, and anxiety/anger EE. EE has also been evaluated using the Emotional Eating Subscale of the Dutch Eating Behavior Questionnaire (DEBQ) [41], a 13-item questionnaire assessing eating behaviors in response to emotions, and using the Emotional Overeating Questionnaire (EOQ) [42,43], a brief 6-item self-report that assesses the frequency of emotional overeating in response to 5 negative emotions (i.e., anxiety, sadness, loneliness, tiredness, and anger) and one positive (happiness).

One of the most common used psychological tools for assessing eating behavior is the Three Factor Eating Questionnaire (TFEQ) [44], which explores three dimensions of eating behavior: uncontrolled eating, emotional eating and cognitive restraint. The TFEQ has showed its clinical utility in a large sample of women with obesity undergoing weight loss treatment [45]. However, this questionnaire was very long (51 items) and this led to the development of a 18 items version, the Three-Factor Eating Questionnaire Revised (TFEQ-R18) [19] and subsequently a 21 items version (TFEQ-R21) [46]. Recently, a short (10-item) questionnaire was developed: the Salzburg Stress Eating Scale (SSES) [47], a psychometric attempt to assess of “pure” stress eating, not only eating related to a specific form of emotion. The SSES has been validated in German population and was demonstrated to be a one-factorial, internally consistent measure, showing moderate correlation with EE [47].

All these questionnaires are similar in design. They all contain items that evaluate the desire for or frequency of food intake in response (all or most items) to negative emotions. The questions are responded to on a Likert Scale, ranging (more or less) from never to very often. The underlying assumption of all these questionnaires is that overeating appears when experiencing negative emotions [4]. For example: “When I feel anxious, I find myself eating” (from the TFEQ-R18). These instruments do not include alternative interpretations of high self-reported emotional eating, as positive emotions [5,16,48], lack of control [49] or eating concerns [15], that have shown their relationship to EE. Alternatively, it has been proposed that self-reported measures eating may reflect beliefs about EE eating rather than one’s actual eating behavior when being emotional [16].

Furthermore, some of these questionnaires are designed to assess eating behavior or eating disorders, and, what is more important, very few validation studies have been conducted to determine the reliability and validity of each instrument across different populations (clinical and non-clinical samples, people with overweight or with obesity and normal weight individuals, different cultures or countries, or groups of different sex and age) [19].

Some of these questionnaires have been translated, adapted, and validated in a Spanish population: validation and psychometric analysis of the TEFQ-R18 and the TEFQ-R21 have been carried out with Spanish students (TEFQ-R18 ages ranging from 12–27 [50]; TEFQ21 ranging from 8.8 to 16.8 [51]). In the same way, the dimensional validity of the DEBQ was reproduced in a female Spanish sample (with ages between 18 and 65 years) [52]. In this study, the Spanish DEBQ was translated from the English version by the authors, who recommended in their validation minor modifications of the questionnaire, like removing item 3. Differences between this and a similar validation study conducted on German participants [53] were found. For instance, item 28 (“desire to eat when bored”) led to inconsistent result patterns in Spanish and German samples. These discrepancies could be attributed to differences between the samples, cultures, or may result from the translation. During the process of questionnaire translation and validation, the original readability of the items should be maintained [54].

Less than a decade ago, it was constructed the Emotional Eater Questionnaire (EEQ), the first EE psychometric measure developed in the Spanish language. The EEQ has a 4-point Likert format (never, sometimes, generally, and always), and its original validation and psychometric analysis were conducted on a Spanish population with obesity [55]. The EEQ includes items about alternative interpretations of high self-reported emotional eating, as lack of control, and does not focus exclusively on negative emotions.

The EEQ has been also used to assess EE in a Spanish population with binge eating disorder [56], and has been validated in Chilean university students, showing its usefulness and its good psychometric properties [57]. Recently, it has been used to show the negative effects of the emotional eating on the physical, psychological and mental in health professional basketball players [58], among other studies.

The original psychometric study of the EEQ identified three factors: disinhibition (lack of control in eating in response to a variety of circumstances, cognitive, emotional or environmental cues), high calorie food preference (type of food that most frequently eaten in given situations) and feelings of guilt (sense of guilt felt by individuals when they look at the weighing scales or when they eat forbidden foods). Additionally, the test provides a global score of EE [55]. However, the psychometric assessment, as well as the analysis of the internal structure, test-retest reliability and convergent validity of this tool were conducted in 354 participants with obesity, all of them in dietetic and behavioral treatment to lose weight. There is no record of analysis or validation of this instrument for other Spanish populations. However, it seems necessary to examine factor structure, internal consistency, and construct validity of the EEQ in non-clinical samples. Its implementation in the general population will allow to deepen the mechanisms involved in EE, in order to design and develop obesity prevention strategies. It seems particularly interesting to validate this instrument and to analyze its psychometric properties in a college population. Previous research suggests the need to improve the diet and eating habits of university students [18,59], and the relationship between academic performance and EE in adolescents [39].

The aims of this study are (i) to explore the factor structure of the EEQ in Spanish college population, and (ii) to determine its psychometric properties (including reliability, concurrent and convergent validity).

## 2. Materials and Methods

### 2.1. Participants

Sample was composed by 295 students attending a Spanish university. The students ranged from first to third academic year. The age of the participants ranged from 17 to 49 years old (*M* = 21.3 ± 4.49). The BMI [weight (kg)/(height)^2^ (m)] reported showed scored between 13.7 and 39.67 kg/m^2^ (*M* = 22.15; *SD* = 3.240.) (only one participant showed underweight but did not report any diagnosis of psychological problem related to eating behavior or being in psychological treatment). Table 1 shows information about the sample studied. Not having any diagnosis of psychological problem or being in psychological treatment was used as inclusion criteria.

### 2.2. Measures

Participants completed self-reported sociodemographic information on their age, course, nationality, height, weight, eating disorder diagnosis or psychological treatment for any psychological problem. Later, participants fulfilled:
-Emotional Eater Questionnaire (EEQ): with the aim to explore its factor structure and to determine its psychometric properties, EEQ was applied to the participants. This questionnaire consists of 10 items on a 4-category Likert scale from 0 (never) to 3 (always). The original validation with a sample of people with obesity [55] found three subscales: lack of control in eating, high calorie food preference, and feelings of guilt. Additionally, the test provides a global score of EE. The temporal stability shows medium-high correlation in the test-retest average (r = 0.702; *p* < 0.001) and the internal consistency of the subscales range from α = 0.61 to α = 0.77, what indicated the appropriate reliability of the instrument.-State-Trait Anxiety Inventory (STAI): to measure levels of anxiety, the state subscale of the STAI from the inventory validated in Spanish population was used [60]. This subscale is composed of 20 items which evaluate anxiety as a state (a transitory emotional response) and uses a Likert scale of four categories from 0 (almost never) to 3 (very often/almost always). The analysis of psychometric properties of the instrument for university population showed a high internal consistency of state subscale (α = 0.96).-Health habits report: given the importance of adequate eating patterns (which excludes EE) in adopting a healthy lifestyle, it was included four items created ad hoc to assess: (1) general lifestyle (“In general, do you have a healthy lifestyle?”); (2) healthy food consumption (“Do you think that your diet is good?”); (3) awareness about food intake (“Are you aware of the calories you eat?”); and (4) perception about weight (“You consider your weight as very thin/thin/normal/slightly overweight/overweight”). These questions are based on a Likert scale of 3 categories from 0 (no) to 3 (yes).

### 2.3. Procedure

The participants were recruited in person by professors, which collaborated with the present study. Questionnaires were provided after informed consent physically. Participants did not obtain any compensation, making participation totally voluntary and unconditionally. For the only minor participant, permission to participate in the study was requested from their parents. The completion of the questionnaires was carried out during the 2019–2020 academic year and it took around 30 min. The study fully complied with the Helsinki Declaration and was approved by the Research Ethics Committee of the University (17/2019). In the last place, data was uploaded onto a database for analysis using SPSS^®^ software version 22 (SPSS Inc., Chicago, IL, USA) and Factor^®^ software for Windows version 10.10.03 (Universitat Rovira i Virgili, Tarragona, Spain).

### 2.4. Data Analysis

First, it was studied items distribution in the sample (descriptive analysis) and corrected correlations item-test. Second, it was analyzed the internal structure of the scale. To this regard, exploratory factor analysis (EFA) was carried out to explore a simpler structure of the construct. Given that EEQ is based on Likert scale with four levels, EFA was carried out by using polychoric correlation matrix. Bartlett’s statistic and Kaiser-Meyer-Olkin (KMO) test were used to test if factorial model is applicable to our data. After that, unweight least squares (ULS) as extraction method and oblique rotation were used. To determine the factor number, factor eigenvalues above 1.0, parallel analysis based on minimum rank factor analysis [61], and goodness of fit statistics were the criteria followed.

Third, it was studied internal consistency of the questionnaire by Cronbach’s alpha. Evidence of convergent validity was analyzed using Spearman’s correlation of the EEQ with STAI questionnaire, given the relationship between anxiety and emotional eating reported by previous researches [29,30,62]. Specifically, state-anxiety subscale was used, since recent research carried out with a university population has found that levels of state-anxiety show a direct linear association with all subscales of the EEQ, while for trait-anxiety this is only the case in the subscales of high calorie food preference and lack of control [18]. Finally, due to the predictive potential of emotional eating on eating problems, evidence of concurrent validity was calculated using Spearman correlations of EEQ with body mass index (BMI) and with some items about healthy habits.

## 3. Results

### 3.1. Items’ Descriptive Analysis

Table 2 shows descriptive analysis of all the items and correlation of every item with the rest of the test. Means range observed was from 0.254 to 1.369 (score range: 0 to 3). Participants showed lowest score on item 9 (“When you overeat while on a diet, do you give up and start eating without control, particularly food that you think is fattening”? and highest on item 2 (“Do you crave specific foods?”). 

All the items showed positive asymmetry. 6 items showed leptokurtic distribution (items 2, 3, 4, 8, 9, and 10) while four showed platykurtic distribution (item 1, 5, 6, and 7). Regarding item contribution to the internal consistency, item 4 (“Do you have problems controlling the amount of certain types of food you eat?”) and 10 (“How often do you feel that food controls you, rather than you controlling food?”) were the most related to the construct.

### 3.2. Exploratory Factor Analysis

Bartlett’s statistics and KMO tests showed the adequacy of the polychoric correlation matrix to the factor model (χ^2^(45) = 1057.4; *p* < 0.001; KMO = 0.835). Table 3 shows the factor extraction by ULS method based on eigenvalues, what considered two factors as most appropriate model and accounts for 56% of the variance (43% for first factor, and 13% for second one). In this regard, Table 4 shows the rotated loading matrix by using Oblimin rotation. 

Factor weights suggested a first factor composed of seven items (2, 3, 4, 5, 6, and 9), and a second factor composed of three items (1, 7, and 10). As shown in Table 5, this factor structure showed optimal values of goodness of fit statistics (it is considered a fit model if Chi-square *p* > 0.05; RMSEA < 0.06; GFI and CFI ≥ 0.90). In this bifactorial structure, correlation between factors was *r* = 0.532.

However, parallel analysis suggested only one factor by using 500 random correlation and throughout permutation of the raw data [63]. As shown in Table 4, all the items showed factor weights greater than 0.30 in the only factor extracted. Nevertheless, the goodness of fit of this one-dimensional structure was worse than the previous one based on RMSEA and Chi-square index (RMSEA = 0.089 and χ^2^
*p* < 0.001). This one-dimensional structure accounted for 43% of the variance.

Finally, a last one-factor structure after removing item 1, 7 and 10 was carried out based on two criteria: (1) the suggestion of some studies that a minimum of four items is necessary to consider a factor [64], and (2) the content of the items, which we not consider relevant or exclusive, from a theoretical view, to measure emotional eater behavior. This last factor structure showed adequate goodness of fit in all the index, as shows Table 3, and accounted for 57% of the variance.

Reliability analysis, convergent and concurrent validity

Cronbach’s Alpha of the one-dimension test based on 7 items was 0.753. Table 6 shows the Spearman’s correlation of EEQ with STAI, BMI, and items about healthy habits. Correlation of EEQ with anxiety state measure was statistically significant, as well as for participants reports about their lifestyle and healthy food consumption. No relationship was found of EEQ with BMI, awareness about food intake and perception about the own weight.

## 4. Discussion

The aims of the present work were to analyze the factor structure and psychometric properties of EEQ in a Spanish university student population. EEQ is a questionnaire used to assess EE, short and easy to administer, which psychometric assessment, as well as the analysis of the internal structure, had been conducted in a Spanish clinical sample, individuals with obesity in treatment to lose weight [55]. The EEQ has been also validated in Chilean university students, showing its usefulness and its good properties [57]. It was necessary to examine factor structure, reliability, and concurrent and convergent validity of the EEQ in Spanish non-clinical samples, and, specifically, in university populations. It has been described poor eating habits among university students [18,59], and it has been found a significant relation between academic performance and EE at this stage [39]. Obesity prevention is a number one public health research priority, and it is clear that EE may play a significant role in the etiology of obesity in adolescents [65].

The factorial structure proposed in the initial validation study [55], carried out with 354 participants with obesity, identified three factors that explained 60% of the total variance: the first factor (6 items) included questions related with disinhibition in eating; the second one (two items) included questions related with food preference; and the third factor (two items) included questions related with the sense of guilt after eating inappropriate or forbidden foods (e.g., sweets or snacks). However, it has been suggested that it is not advisable to extract factors from a questionnaire with less than four items, in fact, it is enough for the factorial structure to explain around 40% (34).

In the present study, conducted in 295 college students, variables were not distributed normally, which prevented the use of extraction methods based on normal distribution. All items showed good homogeneity rates, which represent a good measure of the construct in question. Following Kaiser’s rule, two factors were obtained. The first factor, that accounted for 43% of the variance, included items 2, 3, 4, 5, 6, 8 y 9, and seems to reflect those behaviors most directly related to EE, according to previous research: food intake triggered by emotions [1] (items 5, 6 and 8), the preference for certain foods [14] (items 2, 3, 4, 6 and 9) and the difficulty in controlling eating [49] (items 2,3,4,8 and 9). EE not only involves eating in response to certain emotions, but also it has been suggested the implication of some executive mechanisms, like impulsivity and the inability to modulate emotional responses [66]. Besides, EE is related to unhealthy food choices [12]. Item 5, that refers specifically to eating that is triggered by emotions, (“Do you eat when you are stressed, angry or bored?”) was the one that showed the most to do with this factor.

The second factor, that accounted for 13% of the variance (items 1, 7 and 10, Item 1: Do the weight scales have a great power over you? Can they change your mood? Item 7: Do you feel guilty when eat “forbidden” foods, like sweets or snacks? Item 10: How often do you feel that food controls you, rather that you control food?), includes questions that do not refer to the EE behavior itself, but the emotional consequences of the lack of control over food intake: sense or feelings of guilt when looking at the weighing scales, eating forbidden foods or having the perception of being controlled by food. In fact, two of these three items (7 and 10) shaped the “feeling of guilt” dimension of the three-factors structure found in the initial validation of the questionnaire, and it was the factor that least explained the variance (16%). This factor, “feeling of guilt”, was considered by the authors of the EEQ to be a clinically interesting dimension for the prevention of binge eating disorder [47]. This dimension perhaps make sense only when assessing a clinical population.

In non-clinical populations, the feeling of guilt does not seem necessarily to be linked to EE: in a study carried out with a sample of college students from the United States it was found that the guilt was not the first feeling triggered by EE in females, and in males the post-eating guilt was not very intense [67]. Among Spanish university students similar results have been found: a recent study showed that females frequently experience feelings of guilt following EE. However, males were unlikely to feel guilt after an EE episode [18]. In that study, the variable sex resulted the most predictive variable the for the subscale guilt, what indicates the existence of other factors associated with this subscale beyond EE.

In short, the three items (1, 7 and 10) of the second factor found in this factor structure analysis seems to evaluate other behaviors and attitudes of a more problematic nature, more related to the clinical population. Despite the theoretical controversy about whether these three items may be less representative in a non-clinical population, goodness-of-fit ratings were appropriate for this two-factor model. However, the recommendation of parallel analysis, a more suitable method for factor selection, was followed and an appropriate adjustment utilized [63]. Given the criticisms of self-values as a method of selecting number of factors [64], a parallel analysis suggested the removing of these three items and the use of a single dimension. This last one-factor structure showed the adequate goodness of fit in all the indexes and accounted for 57% of the variance. Item 9 was the most representative of the construct.

The internal consistency of this one-factor model was good (Cronbach’s α = 0.753) according to the literature [68]. This value is also good compared to that obtained by the authors in two of the three factors of the proposed factor structure. Regarding the convergent validity, a direct relationship between the anxiety-state and the score in 7-items EEQ is found, which indicates the coexistence of these two constructs. Anxiety has shown a great effect on eating behavior [31,32] and has been proposed as a primary trigger for EE episodes [29]. High levels of anxiety or perceived stress have been found not only in people with obesity and with EE in weight loss treatment [69,70], but also in samples of college students [71]. Episodes of EE has been reported as the common response to anxiety [72], and it has been found a direct relation between anxiety and EE [18], and between anxiety and sweet craving (which has been associated with EE) [73] in a university population.

As regards concurrent validity of this 7-items model, the EEQ score was predictive of the perception of healthy lifestyle and healthy food consumption. Thus, the EEQ score predicted 5% and 6.7% respectively the criteria. In a convergent way, it has been found that a healthy lifestyle could benefits on emotional regulation [74]. Eating habits have proven to be an important predictor of eating behavior [75,76,77], and having a family meal and eating in a structured setting was associated with less EE and more food enjoyment [78].

Finally, 7-items EEQ score was not predictive of BMI. Although previous studies have shown a relationship between EE and increased BMI in overweight population in treatment for weight loss [79], and EE was shown to be a strong psychological predictor of weight change, the association between BMI change and emotional eating among general population was attenuate by others factors, like the practice of sport [25] or the impulsivity [80]. In a similar way, among university students, the relationship found between EE and BMI has normally been modulated by other variables, such as depression [81,82], sex [83] or physical activity [84]. Moreover, EE eating was much more strongly associated with overeating in the clinical than in the adolescent population [2]. This would explain the lack of predictive capacity of the score obtained in the EEQ on the BMI.

In short, this study presents a new validation of the EEQ, resulting in a seven items one-factor instrument, easy to understand and to administrate to large samples of non-clinical people, valid and reliable in evaluating the degree of emotion in relation with food intake in a university population. Research has found emotional continuity and stability of EE from the childhood to adulthood [85]. Therefore, identifying early this pattern of food intake in youth can help develop prevention strategies and the seven items one-factor EEQ seems a good tool for early identification of this eating pattern in adolescents and young people. The remaining three items could be considered additional items, very useful to be applied in clinical populations (eating disorders, overweight or people with obesity), that would assess problematic emotional implications consequence of EE.

The study of prevalence and frequency of EE during the formative years at university is necessary, because college students are an important group to consider for interventions targeting obesity prevention. Unhealthy behaviors acquired at this age may persist into adulthood [86]. Besides, the transition from the school to the university can be a stressful time for young adults, which make them more susceptible to weight change [87]. However, as a prospective, it would be interesting to validate this questionnaire in other non-clinical samples of the Spanish population.

This study has several limitations. First, a larger sample would have been desirable. Second, the sample is not heterogeneous regarding to gender, age, and BMI score, something that would have been desirable to avoid possible bias in the results. Third, the lack of standardized tests to account for the concurrent validity of the EEQ. Fourth, a comparative study about EEQ between clinical and non-clinical population would be needed, and a greater exploration across Spanish non-clinical samples would also be beneficial, since models of EE could differ between student and general population samples [66]. Finally, although there is a large body of research and empirical studies about EE, both on clinical samples and normal population, this construct is not as simple as is often assumed. There is a lack of solid theoretical frameworks about EE behavior and about the mechanisms involved (psychological, cognitive, affective, neuropsychological), that difficult to interpret results. For some authors, the exact nature of emotional eating remains elusive [88].

Despite these limitations, the present study represents a new step in the study of the EE phenomenon, which must be improved with future research that complements EFA with confirmatory analysis factor. This type of study would allow one to obtain more solid conclusions about the factor structure of the EEQ, to test whether the proposed structure is replicated in another sample.

## 5. Conclusions

The aims of the present study were to analyze the factor structure and psychometric properties of EEQ in a Spanish university student population. The original validation of the 10-items EEQ conducted with a clinical sample (people with obesity in treatment to lose weight) found three factors structure (control in eating, high calorie food preference, and feelings of guilt). In the present study this model was not replicated. The validation carried out in university students found a one-factor structure (removing three items from the questionnaire), with adequate goodness of fit in all the indexes, that accounted for 57% of the variance. This model shows a good internal consistency and convergent validity, and it seems a good instrument to assess EE in non-clinical populations, specifically among Spanish university students.

## Figures and Tables

**Table 1 ijerph-17-09090-t001:** Description of the sample studied.

Variable	N	Percentage
Gender		
Male	78	26.4%
Female	217	73.6%
Practice sports regularly (daily)		
Yes	201	68.81%
No	94	31.86%
Degree		
Nursing	143	48.5%
Physiotherapy	81	27.5%
Psychology	71	24.1%

**Table 2 ijerph-17-09090-t002:** Items descriptive analysis and item-test correlations.

Item	Mean	Variance	Skewness	Kurtosis	Item-Test Correlation
Item 1	0.847	0.803	0.876	−0.005	0.448
Item 2	1.369	0.485	0.809	0.364	0.425
Item 3	0.667	0.651	1.223	1.152	0.461
Item 4	0.762	0.529	0.718	0.288	0.527
Item 5	1.311	0.776	0.328	−0.527	0.467
Item 6	1.064	0.748	0.479	−0.420	0.444
Item 7	1.037	0.975	0.672	−0.560	0.472
Item 8	0.752	0.765	0.964	0.087	0.472
Item 9	0.254	0.306	2.339	5.553	0.499
Item 10	0.586	0.631	1.332	1.254	0.555

**Table 3 ijerph-17-09090-t003:** Factor structure extracted based on eigenvalues by using USL.

Factors	Eigenvalues	Proportion of Variance	Cumulative Proportion of the Variance
1	4.341	0.434	0.434
2	1.287	0.128	0.562
3	0.893	0.089	
4	0.707	0.070	
5	0.622	0.062	
6	0.567	0.056	
7	0.528	0.052	
8	0.464	0.046	
9	0.324	0.032	
10	0.261	0.026	

**Table 4 ijerph-17-09090-t004:** Exploratory analysis of EEQ and communality corresponding to one and two factor structure (N = 295).

EEQ Items	Two Factors			One Factor ^a^		One Factor ^b^	
	Factor 1	Factor 2	h^2^	Factor 1	h^2^	Factor 1	h^2^
1. Do the weight scales have a great power over you? Can they change your mood?	−0.049	0.817	0.627	0.560	0.314	-	-
2. Do you crave specific foods?	0.569	0.000	0.323	0.532	0.283	0.554	0.307
3. Is it difficult for you to stop eating sweet things, especially chocolate?	0.604	0.027	0.383	0.588	0.346	0.618	0.382
4. Do you have problems controlling the amount of certain types of food you eat?	0.630	0.065	0.444	0.643	0.413	0.668	0.447
5. Do you eat when you are stressed, angry or bored?	0.690	−0.100	0.413	0.557	0.311	0.639	0.409
6. Do you eat more of your favorite food and with less control when you are alone?	0.667	−0.091	0.388	0.545	0.297	0.615	0.378
7. Do you feel guilty when eat “forbidden” foods, like sweets or snacks?	0.004	0.764	0.588	0.576	0.332	-	-
8. Do you feel less control over your diet when you are tired after work at night?	0.527	0.101	0.344	0.578	0.334	0.600	0.360
9. When you overeat while on a diet, do you give up and start eating without control, particularly food that you think is fattening?	0.459	0.386	0.548	0.747	0.558	0.647	0.419
10. How often do you feel that food controls you, rather that you control food?	0.382	0.476	0.565	0.740	0.548	-	-

Note: ^a^ One-factor structure with all the items of the EEQ; ^b^ One-factor structure after removed item 1, 7, and 10.

**Table 5 ijerph-17-09090-t005:** Goodness of fit index for the different factor structures.

	RMSA	χ ^2^	CFI	GFI
Two factors	0.049 (*p* =0.968)	31.775 (*p* = 0.200)	0.990	0.984
One factor ^a^	0.089 (*p* = 0.985)	104.476 (*p* < 0.001)	0.957	0.961
One factor ^b^	0.052 (*p* = 0.907)	21.687 (*p* = 0.088)	0.988	0.983

Note: ^a^ One-factor structure with all the items of the EEQ; ^b^ One-factor structure after removed item 1,7, and 10; RMSA = root mean square error approximation; CFI = comparative fit index; GFI = adjusted goodness of fit index; RMSR = root mean square of residuals.

**Table 6 ijerph-17-09090-t006:** Spearman correlation coefficients between EEQ and measures used for convergent and concurrent validity.

Measures	Spearman Coefficient	*p Value*
STAI ^a^	0.317	<0.001
BMI ^b^	−0.030	0.142
Healthy lifestyle report ^b^	−0.224	<0.001
Healthy food consumption report ^b^	−0.259	<0.001
Awareness about food intake report ^b^	−0.086	0.142
Perception about weight ^b^	−0.113	0.054

Note: ^a^ Convergent validity; ^b^ Concurrent validity; STAI = state-anxiety inventory; BMI = Body mass index.

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
