# Peer review of "Factor Structure and Psychometric Properties of Emotional Eater Questionnaire (EEQ) in Spanish Colleges"

_ijerph, 2020, doi:10.3390/ijerph17239090_

Round 1
Reviewer 1 Report
Congratulations on the job. It is almost complete, but I recommend a few questions: - Delete the space on line 115. - Make sure that the means and standard deviations appear in the text like this: 21.3 ± 4.49. - Better that in table 4 no value appears in bold type - Make sure that in all the tables in the footer the meaning of all the abbreviations appears as for example in table 6: BMI or STAI. - The bibliography is not correct in terms of the norms of the journal.Author Response
We found the reviewer’s comments extremely helpful and we have made changes in the manuscript according to them (highlighted in red).
- We have corrected some format mistakes and format suggestions pointed out by the reviewer.
- We have included the meaning of some abbreviation in the note of all the tables.
- The bibliography is now in terms of the norms of the journal.
Reviewer 2 Report
Thanks for the author provide a questionnaire discussing about measurements of adults’ emotional eater. The topic is interesting and contributive to the measurements of adults’ emotional eater with a developed 10 items emotional eater questionnaire (EEQ) for prevention of obesity.
Major concerns:
- Based on my knowledge, exploratory factor analysis (EFA) and confirmatory factor analysis (CFA) are all very important process when developing the questionnaire. It is still unclear that only undertake EFA rather than CFA of EEQ in this study. Please provide a paragraph about CFA.
- It’s a pity only perform a questionnaire measurement rather further lab test of 295 students. Would you plan to perform a lab test analysis to add readability? Please provide a paragraph about related lab test in Discussion section.
Author Response
We found the reviewer’s comments extremely helpful and we have made changes in the manuscript according to them (highlighted in red)
- Regarding to the CFA, our aim is not to develop a questionnaire, but to explore the factorial structure in our sample. To clarify this aspect, we have reformulated the aim of our study to understand better the only use of EFA. We also remarked it in the data analysis section. However, we have included it in the discussion section as a limitation, and the mention to the need of carrying out a CFA study to confirm the structure proposed.
- In regard to the need of performing a lab test to add readability, we are not sure if we understand what you mean. According to our knowledge, lab test to analyze readability is necessary when the items of a new questionnaire are written. This task is intended to ensure understanding of the content of the items. Nevertheless, in our study we use the EEQ, a tool that is already developed, and adapted and validated in Spanish sample. In fact, it is a questionnaire that has been used in several studies (e.g. Garaulet et al., 2012; Escandón et al., 2018; López Gimera . However, we would appreciate if you want to clarify more about this aspect.
Reviewer 3 Report
The authors made all suggested changes. The manuscript can be accepted.
Author Response
We found the reviewer’s comments extremely helpful and we have made changes in the manuscript according to them (highlighted in red).
We want to thank him/her for the review work which has helped to improve the quality of the manuscript.
Reviewer 4 Report
Thank you for addressing the points raised by the reviewers. As you mention throughout your responses that you are currently collecting data that may address some of the limitations of the current study, it may be worth combining the two sets of data into a two study article which may strengthen the article. For example, it is quite common for a validation study to provide results of a EFA and a CPA. As you have rightly pointed out, these should be conducted in separate samples. If data collection for the CPA is underway, these should be presented together.
Intro
Page 1, Lines 30-38. The definition of emotional eating does not include recognition of positive emotions and there is a growing body of evidence surrounding this. The second sentence does not make sense and uncontrolled eating is not described or its relevance introduced. Finally, the reference to emotional regulation does not fit here as paragraph 2 then goes on to described emotional eating in more detail.
Page 2, Lines 48-49. Again, reference to emotion regulation seems off here as emotional regulation has not been fully addressed.
Page 2, Lines 76-96. Attempts have been made to strengthen this aspect but the rationale for the need for a new scale of emotional eating needs to be developed further still - this is a major pitfall of the current study. The article even states the TFEQ is a validated measure for the Spanish population. There is no reference to the Salzburg emotional eating scale (Meule et al., 2018) or the updated version of the TFEQ. Statements like these scales are too complicated for the general population and are designed to assess eating disorders need to be justified and supported. Some of the criticisms of the previous scales, e.g. positive emotions, are not addressed in the new scale either. The last sentence is difficult to understand and follow.
Page 3, Line 104-105. Greater description is needed on what the EEQ is measuring. How does disinhibition or lack of control differ from uncontrolled eating for example.
Method
Page 3, Line 124. Thank you to the authors for providing further information on these two participants (BMI and age). A further query would be that if only one participant fell into the underweight category it could be possible, they are an outlier. If a BMI value of 18.5 and below is an underweight classification, this would mean that participant is almost 5 BMI points below the other participants. Similarly, if only one participant is a minor then they could also be an outlier as they are the only participant considered a minor and eating behaviours may differ for them, e.g. if they still live at home with parents.
Table 1 – What does “regular practice” refer to?
Page 4 – EEQ. There are a number of missing words and grammatical errors that make this section hard to follow
Page 4 – Data analysis. There are a number of missing words and grammatical errors that make this section hard to follow
Discussion
Page 7, Lines 228-230. This sentence does not make sense
Page 7/8, Lines 248-254. It states here that the scale assessed variables that have been related to emotional eating, this would indicate that the scale is not measuring emotional eating, either in the desire/urge to eat when experiencing emotions or recalling the increase/decrease in emotions. Instead it reflects factors that might increase vulnerability. Indeed, item 5 had the strongest loading and this is in relation to specific emotions and whether participants eat in relation to emotions.
Page 8, Lines 255-258. It states here that questions 1, 7 and 10 do not refer to emotional eating. It could be argued that 2, 3, 4, and 9 also do not refer to emotional eating behaviours.
Page 9, Lines 318-319. What is meant by the lack of theoretical frameworks? There is a large body of research on the theories of emotional eating. Overall, the limitations section is still weak and merely lists limitations
References
Meule, A., Reichenberger, J., & Blechert, J. (2018). Development and preliminary validation of the Salzburg Stress Eating Scale. Appetite, 120, 442-448.
Author Response
We found the reviewer’s comments extremely helpful and we have made changes in the manuscript according to them (highlighted in red).
INTRO:
- We have included references about EE in response to positive emotions. Thank you for pointing.
- We meant “emotional eating” instead of “uncontrolled eating”. We have changed the sentence.
- We have further developed the mechanisms related to EE (emotional dysregulation) and clarified the paragraph. And we have added about emotional dysregulation in the paragraph that talks about intervention in people with obesity
- We have further justified the need for the study: the limitations of other instruments and the importance of developing and validating a tool in Spanish for the Spanish population.
- We have included the reference and some explanation about Salzburg Stress Eating Scale (SSES)
- The TFEQ has been validated in a Spanish population, but it is not a specific measure for EE. We have added this explanation.
- We have justified with a reference the statement “these scales are too long”.
- We have given a greater description on what the EEQ is measuring (disinhibition, food preference and feelings of guilt).
METHOD
- We agree with the reviewer about the results would be more robust if EFA and CFA is presented together. We have added it to the discussion as a limitation and as prospective. However, we are interested in a first exploratory study, to describe the factor structure of the EEQ in our sample, previous to a future CFA study to confirm the structure found. To this regard, there are some published manuscript with the same purpose that the authors only carry out EFA (e.g. Garaulet et al., 2012). However, we hope to have data soon to carry out CFA, to discuss the results of the present study.
- Regarding to the 4% of the sample who do not live with their parents, they live at the university residence. For this reason, we consider that the eating pattern of these participants does not depend on them, as in the case of the 96% of the sample, who live with their parents. Regard to the participant with low BMI, he can be considered as an outlier, however, we consider it is not relevant to our results for different reasons:
- We checked that the participant had no diagnosis of psychological problem or was under psychological treatment. Thus, his BMI is irrelevant to measure his eating patterns.
- Despite BMI is related to some eating problems, it is not always a good indicator of mental health. In fact, it is every day a more questionable index (Wells, 2014).
Nevertheless, to avoid confusion to the reader we have removed the BMI range and we added the heterogeneity of the sample as a limitation in the discussion section.
- We have specified in table 1 what “regular practice” refers to.
- We have corrected missing words and grammatical errors of EEQ instrument description (page 4) and Data analysis section (page 4).
DISCUSSION
- We have improved the wording of some sentence of difficult comprehension.
- We have changed the wording of the paragraph describing the mechanisms involved in the EE that evaluate factor 1 items, and we have added references. We hope to have clarified why we believe that this factor really reflects
- We have we have explained why items 2, 3, 4 y 9 (and the rest of the items of the one-factor scale) refer to emotional eating behavior.
- We think that, despite the large body of research on the theories of emotional eating, it is not clear all the mechanisms involved in EE, in clinical and normal population. We explain this point in the discussion
- We have further developed the section on limitations
Our study presents a new validation of the EEQ, resulting in a 7 items one-factor instrument, valid and reliable and easy to understand and to administrate to large samples of non-clinical people, specifically in college population. We do not underestimate the remaining three items, these three elements, present in the original version of the EEQ, but we consider these items additional items, very useful to be applied in clinical populations. We have added some paragraph about this in de discussion
Sincerely,
Elena Bernabéu Brotons, in behalf of all the authors
Round 2
Reviewer 4 Report
Thank you for addressing the points raised by the reviewers and providing a thorough response. Points raised throughout the manuscript are clearer to the reader.
Abstract
Page 1, Lines 24-26. This study did not assess sub-clinical versus individual’s with obesity so unsure this conclusion is appropriate here
Intro
Page 1, Lines 39-42. This sentence is difficult to follow.
Page 2, Line 71. This sentence is difficult to follow.
Page 2-3. Lines 74-121. The rationale for the need for a new emotional eating remains weak. The limitation that the previous measures are too long does not stand up (TFEQ-18: 3 items, TFEQ-21: 6 items). The limitations that have been listed are also not addressed by the items in the new measure. The previous factors of the EEQ also indicate other reasons for consuming food, e.g. cognitive and environmental, and these have been suggested as limitations of the existing measures. Items that refer to craving foods have no situation attached to them so therefore reflects a general tendency to crave foods but not in a given situation, such as in response to emotions. The reference to Jáuregui-Lobera et al 2014 is confusing as they did find a 3 item emotional eating subscale in their factor analysis. The DEBQ has also been validated in Spanish and the newest version of the TFEQ has as well (Children and adolescents though). These articles also have the strength of presenting both an exploratory and confirmatory analysis.
Cebolla, A., Barrada, J. R., Van Strien, T., Oliver, E., & Baños, R. (2014). Validation of the Dutch Eating Behavior Questionnaire (DEBQ) in a sample of Spanish women. Appetite, 73, 58-64.
Martín-García, M., Vila-Maldonado, S., Rodríguez-Gómez, I., Faya, F. M., Plaza-Carmona, M., Pastor-Vicedo, J. C., & Ara, I. (2016). The Spanish version of the Three Factor Eating Questionnaire-R21 for children and adolescents (TFEQ-R21C): psychometric analysis and relationships with body composition and fitness variables. Physiology & behavior, 165, 350-357.
Method
Page 3, Line 144. Thank you to the authors for further clarifying their thorough investigation into outliers. For transparency, it may be useful to keep the range in the text and add an explanation to the footnotes.
Page 4, Table 1. Sorry for the previous lack of clarity, what does regular practice refer to? Regular practice of what?
Page 4, Lines 166-168. This sentence is difficult to follow.
Discussion
Page 8, Line 251. To ensure non-stigmatising language is used throughout, “individuals with obesity”. The manuscript should be thoroughly checked for this.
Author Response
Attached please find the revised version of manuscript number ijerph-980394 titled: “Factor structure and psychometric properties of Emotional Eater Questionnaire (EEQ) in Spanish colleges”. We want to thank reviewer 4 for his / her work, that has helped to improve the quality of the manuscript. We found his /her comments extremely helpful and we have made changes in the manuscript according to them (highlighted in red).
Abstract
Page 1, Lines 24-26. This study did not assess sub-clinical versus individual’s with obesity so unsure this conclusion is appropriate here
- We have modified this paragraph to reflect properly the findings of the study
Intro
Page 1, Lines 39-42. This sentence is difficult to follow.
- We have changed the phrasing of this paragraph so that it is now much clearer.
Page 2, Line 71. This sentence is difficult to follow.
- We have changed the wording of this paragraph.
Page 2-3. Lines 74-121. The rationale for the need for a new emotional eating remains weak. The limitation that the previous measures are too long does not stand up (TFEQ-18: 3 items, TFEQ-21: 6 items). The limitations that have been listed are also not addressed by the items in the new measure. The previous factors of the EEQ also indicate other reasons for consuming food, e.g. cognitive and environmental, and these have been suggested as limitations of the existing measures. Items that refer to craving foods have no situation attached to them so therefore reflects a general tendency to crave foods but not in a given situation, such as in response to emotions. The reference to Jáuregui-Lobera et al 2014 is confusing as they did find a 3 item emotional eating subscale in their factor analysis. The DEBQ has also been validated in Spanish and the newest version of the TFEQ has as well (Children and adolescents though). These articles also have the strength of presenting both an exploratory and confirmatory analysis.
Cebolla, A., Barrada, J. R., Van Strien, T., Oliver, E., & Baños, R. (2014). Validation of the Dutch Eating Behavior Questionnaire (DEBQ) in a sample of Spanish women. Appetite, 73, 58-64.
Martín-García, M., Vila-Maldonado, S., Rodríguez-Gómez, I., Faya, F. M., Plaza-Carmona, M., Pastor-Vicedo, J. C., & Ara, I. (2016). The Spanish version of the Three Factor Eating Questionnaire-R21 for children and adolescents (TFEQ-R21C): psychometric analysis and relationships with body composition and fitness variables. Physiology & behavior, 165, 350-357.
- Thank you for your suggestions, we have revised this paragraph and the justification for our study. We have modified the limitations of the EE measures and we have indicated some advantages of the EEQ. In fact, we have not developed the EEQ questionnaire, but we have used it in previous studies (as others Spanish researchers), and we thought about the need for the validation study in normal population.
We have referred to the validation studies suggested, and we have simplified and clarify the reference to Jáuregui-Lobera et al.
Method
Page 3, Line 144. Thank you to the authors for further clarifying their thorough investigation into outliers. For transparency, it may be useful to keep the range in the text and add an explanation to the footnotes.
- We have re-indicated the BMI index in the text, and we have added the explanation to the footnotes.
Page 4, Table 1. Sorry for the previous lack of clarity, what does regular practice refer to? Regular practice of what?
- We have specified we refer to sports regular practice.
Page 4, Lines 166-168. This sentence is difficult to follow.
- We have changed the phrasing of this paragraph so that it is now much clearer.
Discussion
Page 8, Line 251. To ensure non-stigmatising language is used throughout, “individuals with obesity”. The manuscript should be thoroughly checked for this.
- We have changed this word (obese) for the expression “people or participants with obesity”. The authors of the different studies distinguish between overweight and obesity, and specifically use the latter term.

This manuscript is a resubmission of an earlier submission. The following is a list of the peer review reports and author responses from that submission.
Round 1
Reviewer 1 Report
- Check the format to cite and reference
- The citations are in brackets and the references are not adapted to the journal's standards
Introduction
- It would be convenient to introduce information about university students and their eating behavior, poor eating habits and the association with academic performance to understand the importance of this study and justify the selection of the sample.
Materials and Methods
Participants:
- It is not a heterogeneous sample in terms of gender, since it was for convenience, this could have been taken into account to avoid possible bias in results.
- Add the inclusion criteria for the selection of the sample.
Measures:
- It would have been good to take into account the sentimental situation, if they have a partner or are single.
Procedure:
- Detail what the students were studying
- What academic year are they in?
Data analysis:
- Detail and develop this section better: specify SPSS version.
Add questionnaire
Reviewer 2 Report
This is a questionnaire discussing about measurements of adults’ emotional eater.
I think the topic is interesting and contributive to the measurements of adults’ emotional eater with a developed 10 items questionnaire for prevention of obesity.
Major concerns:
- Please summarize the outcomes of the previous studies related to questionnaire discussing about measurements of adults’ emotional eater if available.
- Based on my knowledge, exploratory factor analysis (EFA) and confirmatory factor analysis (CFA) are all very important process when developing the questionnaire. It is still unclear that only undertake EFA rather than CFA of emotional eater questionnaire in this study.
- The authors should provide a table summarized characteristics of the population studied.
- Was an external validation of emotional eater questionnaire conducted?
- Please provide percentage of separated factors explained of the total variance.
- Some references are outdated and should be updated accordingly.
- To maintain the quality of IJERPH papers, this article requires major editing by a native English speaker both in language and coherence.
- It’s a pity only perform a questionnaire measurement rather further lab test of 295 students. Would you plan to perform a lab test analysis to add readability?
Reviewer 3 Report
The manuscript is novel and the authors provide very important information regarding the impact of emotional factors on food intake. The manuscript has an adequate structure, the methodology is consistent with the purpose of the study. The results support the argument. However, I have the following comments.
I. Major Comments:
1. In the introduction I suggest including a brief paragraph regarding the intake of i) food, ii) energy, iii) micro and macronutrients, and the impact on nutritional status.
2. In the results, can the description of the participants go in a table?
3. The discussion is good, but I suggest including a paragraph that includes aspects related to food and nutrient intake, and nutritional status. In this regard, it would be interesting for the authors to discuss results related to the quantification of intake (questionnaires of frequency of consumption) and nutritional status.
Suggested references:
The Impact of Maternal Diet during Pregnancy and Lactation on the Fatty Acid Composition of Erythrocytes and Breast Milk of Chilean Women. Nutrients. 2018; 10 (7): 839. PMID: 29958393
Diet, Plasma, Erythrocytes, and Spermatozoa Fatty Acid Composition Changes in Young Vegan Men. Lipids. 2020. PMID: 32757304
II. Minor comments:
1. Improve the writing of the study objective.
Reviewer 4 Report
Appropriate research which thoroughly examines the psychometric properties of measures is important. This study aims to assess the psychometric properties of a novel measure of emotional eating in a sample of Spanish college students. Whilst this aim of this article does fit into the scope of IJERPH, I do not believe this manuscript is suitable for publication. Throughout the manuscript, considerable additional justification and support is needed for arguments presented and there are a number of limitations to the study design. Although these limitations cannot be rectified within the current study they need to be addressed thoroughly in the discussion or alternatively, they could be addressed in a subsequent study which would substantially strengthen the manuscript.
Below I provide a more detailed explanation of what led to my decision.
Abstract:
Line 14 – There are a number of questionnaires which assess emotional eating which are widely used. The argument of why a new measure is needed needs to be strengthened beyond more than a gap in the research.
Line 16 – Should this be State-Trait anxiety inventory?
Line 21-22 – This sentence is unclear as it appears there are some words missing.
Introduction:
The definition of emotional eating and its importance is presented well. A possible additional point to address in terms of self-report measures of emotional eating is the potential for report bias and whether measures are correlated with actual food intake. Some possible articles to explore could be Adriaanse et al (2011), Evers et al. (2009), Evers et al. (2018) and Cardi et al. (2015).
Line 35 – A useful and recent review article to direct readers to could be van Strien (2018)
Line 48-52 – Whilst the relationship between anxiety and emotional eating is an important point to raise this paragraph seems in a strange place and does not link with the previous and subsequent paragraphs. Other factors, for example depression, have also be associated with emotional eating. Is there a reason depression has not been addressed in the current study?
Line 53-70 – There needs to be a stronger argument as to why the previous measures of emotional eating may not be suitable and support for these arguments need to be included. The emotional eating scale, for example, was developed in a population with obesity but a child/adolescent version is available, and a second version has been validated in undergraduate students. Similarly, the Dutch Eating Behaviour Questionnaire was developed through thorough testing of items and across different population samples. It is now widely used in research. The Three Factor Eating Questionnaire has also been extensively researched and revised. The DEBQ emotional eating subscale has a similar number of items as the EEQ and the TFEQ has a similar response scale as the EEQ. Therefore, it is not clear how the EEQ improves on ease of use. The argument also needs to be stronger in terms of the current study. For example, it is stated that questionnaires need to validated across sexes, age groups and weight groups but the current study only assesses the EEQ within Spanish undergraduate students.
Method
Line 89 - The range of ages is stated as being from 17-49 years, was parental consent obtained for those under 18 years old? If so, how was this obtained, and if it is not required it would be useful to include a statement acknowledging this.
Line 90 – Rounding error in the % for women.
Line 91 – The minimum BMI value is very low given that BMI values below 18.5 are categorised as underweight. It may be worth indicating how many participants are classified as underweight and whether participants may have reported errors with their height or weight. Is it possible a small sample may have an undiagnosed eating disorder? Did any participants respond on the question on the eating disorder question that they had a current or historical diagnosis?
Line 97-103 – The EEQ validation paper and Merchena et al (2020) refer translating global scores into categories of being a non- to high- emotional eater. It would be useful to reference this even if the current study no longer suggests these cut-offs are appropriate. This could be another aspect of your study to consider and address. Did participants complete the Spanish or English version of the EEQ? Were all measures in English?
Line 104-110 – Was the state or trait anxiety subscale used, or both? There is only one alpha reported and between this section and the data analysis section it is unclear which subscale was used.
Line 111-115 – What were the response format for these questions? What is the justification of including single items as a form of concurrent validity?
Line 119 – What is meant by participants being ‘disinterested’?
Line 124-141 – This section is rather difficult to follow. There are a few typos and word order issues.
Results
Line 151 – Item 10 is highlight as having being the most related to the construct but then is later removed following the factor analysis. It seems strange to highlight this of importance but to later remove the item as not contributing.
Line 157-187 – This section is difficult to follow. Words appear to be missing throughout.
Discussion
Line 190-192 – It would be clearer to separate this into two sentences addressing the strengths of the EEQ and the previous findings.
Line 201-203 – It appears this sentence is making two points, one about the factors with less than 4 items and a second point about factor structure but it is currently difficult to follow. What impact might these findings have on studies that have used the EEQ in both its global and subscale format, for example Marchena et al (2020).
Line 208 – the statement “theoretically refer to emotional eating” appears to be contradicted later in the discussion when it is stated that there is an ‘absence of solid theoretical frameworks’. It could also be argued that the items are do not refer to emotional eating. Whilst item 5 specifically asked about eating in relation to emotions, the remaining items do not. The proposed unidimensional scale is a combination of items that were previously labelled as lack of control and a preference for high calorie foods. Emotional eating is defined in the introduction as “eating that is triggered” by emotions and “tendency to consume food in order to cope with negative emotional states”. Examining the other items, they appear to be similar to those included in the TFEQ that reflect uncontrolled eating. Indeed, one of the three subscales generated in the initial study (Garaulet et al., 2012) and in Marchena et al. (2020) is of a lack of control in eating. As participants did not complete previously published measures of emotional eating it is difficult to state that emotional eating is being measured here, likewise that they are different to measures of uncontrolled eating or similar. Justification is also needed as to why previous measures have not been included, especially as this study aimed to assess the psychometric properties of the EEQ.
Line 216-221- Greater clarity is needed here. This factor is identified as referring to guilt, but it is unclear how items 1 and 10 relate to guilt. In addition, it is stated that this factor is not necessary for the non-clinical population but then goes on to present a study in which females have been shown to experience guilt after emotional eating.
Line 265 – The limitations section is weak. Justification of why a larger sample would be desirable is needed, likewise, as the study was an investigation of the psychometric properties of the EEQ it seems strange that current standardised tests of emotion have been omitted. In addition to clinical versus non-clinical samples, greater exploration across non-clinical samples would also be beneficial. For example, studies have shown models of emotional eating differ between student and general population samples (e.g. Pink et al., 2019)
References
Adriaanse, M. A., de Ridder, D. T., & Evers, C. (2011). Emotional eating: Eating when emotional or emotional about eating?. Psychology and Health, 26(1), 23-39.
Cardi, V., Leppanen, J., & Treasure, J. (2015). The effects of negative and positive mood induction on eating behaviour: A meta-analysis of laboratory studies in the healthy population and eating and weight disorders. Neuroscience & Biobehavioral Reviews, 57, 299-309.
Evers, C., de Ridder, D. T., & Adriaanse, M. A. (2009). Assessing yourself as an emotional eater: Mission impossible?. Health Psychology, 28(6), 717.
Evers, C., Dingemans, A., Junghans, A. F., & Boevé, A. (2018). Feeling bad or feeling good, does emotion affect your consumption of food? A meta-analysis of the experimental evidence. Neuroscience & Biobehavioral Reviews, 92, 195-208.
Pink, A. E., Lee, M., Price, M., & Williams, C. (2019). A serial mediation model of the relationship between alexithymia and BMI: The role of negative affect, negative urgency and emotional eating. Appetite, 133, 270-278.
Van Strien, T. (2018). Causes of emotional eating and matched treatment of obesity. Current diabetes reports, 18(6), 35.